# Mast cells promote pathology and susceptibility in tuberculosis

Ananya Gupta[1,2†], Vibha Taneja[1†], Javier Rangel-Moreno[3], Nilofer Naqvi[1], Abhimanyu[1], Yun Tao[1], Mushtaq Ahmed[1], Kuldeep Singh Chauhan[1], Daniela Trejo-Ponce de Leon[4,5], Gustavo Ramírez-Martínez[5], Luis Jiménez-Alvarez[5], Cesar Luna-Rivero[5], Joaquin Zuniga[4,5], Deepak Kaushal[6], Shabaana A Khader[1,2]*

[1]Department of Microbiology, The University of Chicago, Chicago, United States; [2]Department of Molecular Microbiology, Washington University in St. Louis, St. Louis, United States; [3]Division of Allergy, Immunology and Rheumatology, Department of Medicine, University of Rochester Medical Center, Rochester, United States; [4]Technologico de Monterrey, Escuela de Medicina y Ciencias de la Salud, Mexico City, Mexico; [5]Laboratory of Immunobiology and Genetics and Department of Pathology, Instituto Nacional de Enfermedades Respiratorias Ismael Cosio Villegas, Mexico City, Mexico; [6]Southwest National Primate Research Center, Texas Biomedical Research Institute, San Antonio, United States

*For correspondence:
khader@uchicago.edu

†These authors contributed
equally to this work

Competing interest: The authors
declare that no competing
interests exist.

Reviewing Editor: Amit Singh,
Indian Institute of Science, India

## eLife Assessment

In this **useful** study, the authors utilize published scRNA-seq data to highlight the potential importance of mast cells (MCs) in TB granulomas, presenting a **solid** comparative assessment of chymase- and tryptase-expressing MCs in the lungs of *Mycobacterium tuberculosis*-infected individuals and non-human primates. While the authors appropriately discussed the inconsistencies across models, adoptive transfer experiments in MC-deficient mice would substantially strengthen the causal link between MCs and TB outcomes, providing more direct functional validation of the proposed role of MCs in TB pathogenesis.

**Abstract** Tuberculosis (TB), caused by the bacterium *Mycobacterium tuberculosis* (*Mtb*), infects approximately one-fourth of the world's population. We reported an increased accumulation of mast cells (MCs) in the lungs of macaques with active pulmonary TB (PTB), compared with those with latent TB infection (LTBI). MCs respond in vitro to *Mtb* exposure via degranulation and by inducing proinflammatory cytokines. In the current study, we demonstrate an increased production of chymase by MCs in granulomas of humans and macaques with PTB. Single-cell (sc) RNA sequencing analysis revealed distinct MC transcriptional programs between LTBI and PTB, with PTB-associated MCs enriched in interferon gamma, oxidative phosphorylation, and MYC signaling. In a mouse model, MC deficiency led to improved control of *Mtb* infection that coincided with reduced accumulation of lung myeloid cells and diminished lung inflammation at chronic stages of infection. Airway transfer of MCs into wild-type *Mtb*-infected mice showed increased neutrophils, decreased recruited macrophages, and elevated *Mtb* dissemination to the spleen. Together, these findings highlight MCs as active drivers of TB pathogenesis and potential targets for host-directed therapies for TB.

## Introduction

Tuberculosis (TB) remains a significant global health issue, with approximately one-quarter of the world's population harboring *Mycobacterium tuberculosis* (*Mtb*), causing around 1.25 million deaths each year (**WHO, 2024**). The disease often starts as a latent TB infection (LTBI), in which the bacteria may remain dormant without disease symptoms. However, LTBI can progress to active pulmonary TB (PTB), characterized by severe respiratory symptoms and high transmission potential. The immune mechanisms that allow progression from latency to PTB are not fully defined. Thus, understanding the immune factors that drive progression toward PTB will allow the development of novel therapeutics for TB control. Toward this overall goal, we recently showed that the lung single-cell transcriptional immune landscape during LTBI and PTB in *Mtb*-infected macaques was distinct. For example, PTB was characterized by the significant accumulation of Type I IFN-expressing plasmacytoid dendritic cells (DCs), IFN-responsive macrophages, as well as activated T cells in the lungs (**Esaulova et al., 2021**). Additionally, mast cells (MCs) were increased in the lungs of macaques with PTB (**Esaulova et al., 2021**). In sharp contrast, LTBI was characterized by increased presence of cytotoxic NK cells but lack of recruitment of MCs in the lungs (**Esaulova et al., 2021**).

MCs are found in the lung where they influence inflammatory responses (**Virk et al., 2016**; **Wasserman, 1984**). MCs have been shown to respond in vitro to *Mtb* exposure via surface receptors such as CD48 (**Muñoz et al., 2003**). They also respond to *Mtb* exposure or mycobacterial lipids by undergoing degranulation of prestored granules, such as histamine and β-hexosaminidase, and secrete proinflammatory cytokines such as IL-6 and TNF-α (**Muñoz et al., 2003**). Degranulation of MCs following intratracheal infection with a high dose of *Mtb* was shown to limit inflammation and the production of proinflammatory cytokines such as IL-1β and TNF-α (**Carlos et al., 2007**). MCs produce and release either chymase or tryptase (**Bian et al., 2021**), which are both proteases that are stored in the cell's secretory granules. Recent studies with lung biopsies of TB patients showed an enrichment of MCs expressing IL-17 at inflammatory sites. In contrast, chymase-rich MCs (MC$_C$s) producing TGF-β were detected in proximity to mature granulomas in lung biopsies from PTB (**Garcia-Rodriguez et al., 2021**). Furthermore, while healthy lung predominantly has tryptase-expressing mast cells (MC$_T$s), both chymase and MCs co-expressing chymase and tryptase (MC$_C$s and MC$_{TC}$s) accumulate in the infected lung of patients with PTB (**Garcia-Rodriguez et al., 2021**). Thus, while previous studies have shown that MCs respond to *Mtb* exposure and accumulate in macaque and human lungs during PTB, it is not completely known if MCs functionally mediate protective or pathological outcomes in the context of TB infection.

In the current study, we showed that the distribution and localization of MCs in PTB in humans and macaques were associated with chymase production. Using scRNA seq analysis, we show that MCs found in LTBI and healthy lungs in macaques are transcriptionally distinct from PTB lungs, showing enrichment of tumor necrosis factor alpha, cholesterol, and transforming growth factor beta signaling. In contrast, MCs found in PTB express increased levels of signatures associated with interferon gamma, oxidative phosphorylation, and MYC signaling. Additionally, mice deficient in MCs showed improved control of *Mtb* infection and reduced lung inflammation. Airway transfer of MCs into wild-type *Mtb*-infected mice increased lung neutrophils and elevated *Mtb* dissemination to the spleen. These results together provide novel evidence that MCs contribute to immune pathology and reduced *Mtb* control, suggesting a pathological role for MCs during *Mtb* infection.

## Results

### MCs localize and transition phenotypes within TB granulomas

A previous study observed that tryptase-expressing MC$_T$s were primarily found within the lungs of healthy controls (HC), while chymase-expressing MC$_C$s or both chymase and tryptase-expressing MC$_{TC}$s were found in the lung of patients with PTB (**Garcia-Rodriguez et al., 2021**). To corroborate these observations and analyze the compartmentalization of MCs in human lungs, we stained lung biopsies from healthy individuals and patients with PTB to visualize the spatial distribution of MC$_T$s, MC$_C$s, and MC$_{TC}$s. Lung granulomas from PTB patients were further classified by the presence or absence of necrosis, as early or late granulomas. Early granulomas had well-defined immune and stromal cell clusters without central necrosis. Late granulomas were larger, with necrotic cores containing bacteria and dead neutrophils, surrounded by lymphocytes. MCs were quantified both within and around

early granulomas, whereas in late granulomas, they were primarily measured in the peripheral regions surrounding the necrotic center. Considering lung parenchyma, interstitium, vasculature, or bronchus, we observed that HC lungs predominantly contain MC$_T$s and, to a lesser extent, MC$_{TC}$s. In TB lesions from PTB patients, MC$_{TC}$s accumulated in early immature granulomas, whereas MC$_C$s accumulated in late granulomas (*Figure 1A, B*). MC$_T$s also increased in the interstitium, vasculature, and bronchus-associated lymphoid tissue of PTB patients (*Figure 1—figure supplement 1A-C*). However, we did not observe any differences in MC$_{TC}$s and MC$_C$s at these sites (*Figure 1—figure supplement 1D–I*). These observations confirmed that tryptase-expressing MC$_T$s are found in HCs (*Garcia-Rodriguez et al., 2021*), while the dual tryptase and chymase-expressing MCs were seen in early granulomas, and only chymase-associated MCs were observed in late granulomas with necrotic cores, defining the MCs associated with TB disease progression.

Our previously published data showed that MCs accumulate in the lungs of macaques with PTB compared to LTBI (*Esaulova et al., 2021*). Thus, we next analyzed the accumulation and localization of MCs in the lungs of macaques with LTBI and PTB. We found that, similar to human healthy lungs, MC$_T$s accumulated in the lungs of healthy macaques. Although MC$_T$s increased in some lesions in the lungs of macaques with LTBI, the numbers of MC$_T$s in macaques with PTB were significantly increased in all sites, including the granuloma (*Figure 1C, D*), interstitium, vasculature, as well as bronchus-associated lymphoid tissue of PTB patients (*Figure 1—figure supplement 1J–L*). Additionally, MC$_{TC}$s were significantly increased within the granulomas of macaques with PTB compared to the lungs of macaques with LTBI and HCs (*Figure 1C, D*), but did not differ at any other sites in the lung (*Figure 1—figure supplement 1M-O*) compared to HCs. However, we did not observe any increase in these cells at other sites within the lung compared to healthy macaques. MC$_C$s were not measurable in any region of the macaque lungs. Our data indicate that during LTBI, there is an accumulation of MC$_T$s but not MC$_{TC}$s. However, as the disease progresses to PTB, both MC$_{TC}$s and additional MC$_T$s are elevated, particularly within the granulomatous lesions.

## Lung single-cell transcriptome in macaques with TB exhibits MC diversity

In the previous section, we found that tryptase protein expression on MCs was lower in LTBI, and as the disease progressed to PTB, MCs expressed chymase, either alone MC$_C$s or in combination with tryptase, MC$_{TC}$. To further validate whether this increase in tryptase and chymase protein expression was also reflected in the single-cell transcriptomes during PTB, LTBI, and HCs, we reanalyzed the MCs from our previously published data from NHPs (*Esaulova et al., 2021*). These macaques were infected with 10 bacilli to generate LTBI, or with 100 bacilli for a progressive PTB infection, or were uninfected (*Figure 2A*). The median duration of infection for the PTB macaques was 10 weeks, and the LTBI macaques underwent necropsy at a median of 23 weeks post-infection. We analyzed the single-cell transcriptomes of 500 MCs using unsupervised clustering and identified four distinct clusters. Three clusters (0, 1, and 3) belonged to the PTB group, while cluster 2 was found exclusively in LTBI and HC, with the majority of MCs coming from the PTB condition (*Figure 2B*). All the MC clusters were positive for canonical markers such as *FCER1A* (high-affinity IgE receptor), *MS4A2* (IgE subunit), *CD48* (MC receptor), and negative for markers like *ITGAX* (macrophage/DC marker) (*Figure 2—figure supplement 1A*), with distinct differentially expressed genes (DEGs) (*Figure 2—figure supplement 1B*). Reactome gene ontology analysis of cluster-specific DEGs revealed enrichment of cholesterol, TNF-α, and TGFβ signaling in LTBI, while oxidative phosphorylation, IFNγ signaling, and MYC signaling were enriched in the PTB group (*Figure 2C*). Plotting the summed *Ucell* module scores revealed significant upregulation of IFNγ signaling, oxidative phosphorylation, and Th2 signature in PTB ($p < 0.05$), while LTBI and HC clusters showed enhanced TNF-α signaling ($p < 0.05$) (*Figure 2D–G*; *Figure 2—figure supplement 1C-F*). These results indicate that MCs exhibited distinct transcriptomic profiles depending on the disease state, with MCs from LTBI and HC showing more metabolically active pathways, while MCs from PTB display a more proinflammatory signature.

Since we observed increased tryptase and chymase in MCs of PTB macaques (*Figure 1C*), we next examined the levels of tryptase and chymase genes within the single-cell dataset. As we were examining these genes across species, we observed considerable variation in sequence similarity and functional annotation of tryptase genes between humans and NHPs. While *TPSG1* (encoding γ tryptase) and *TPSD1* (encoding δ tryptase) share the gene name in humans and NHPs, the gene corresponding

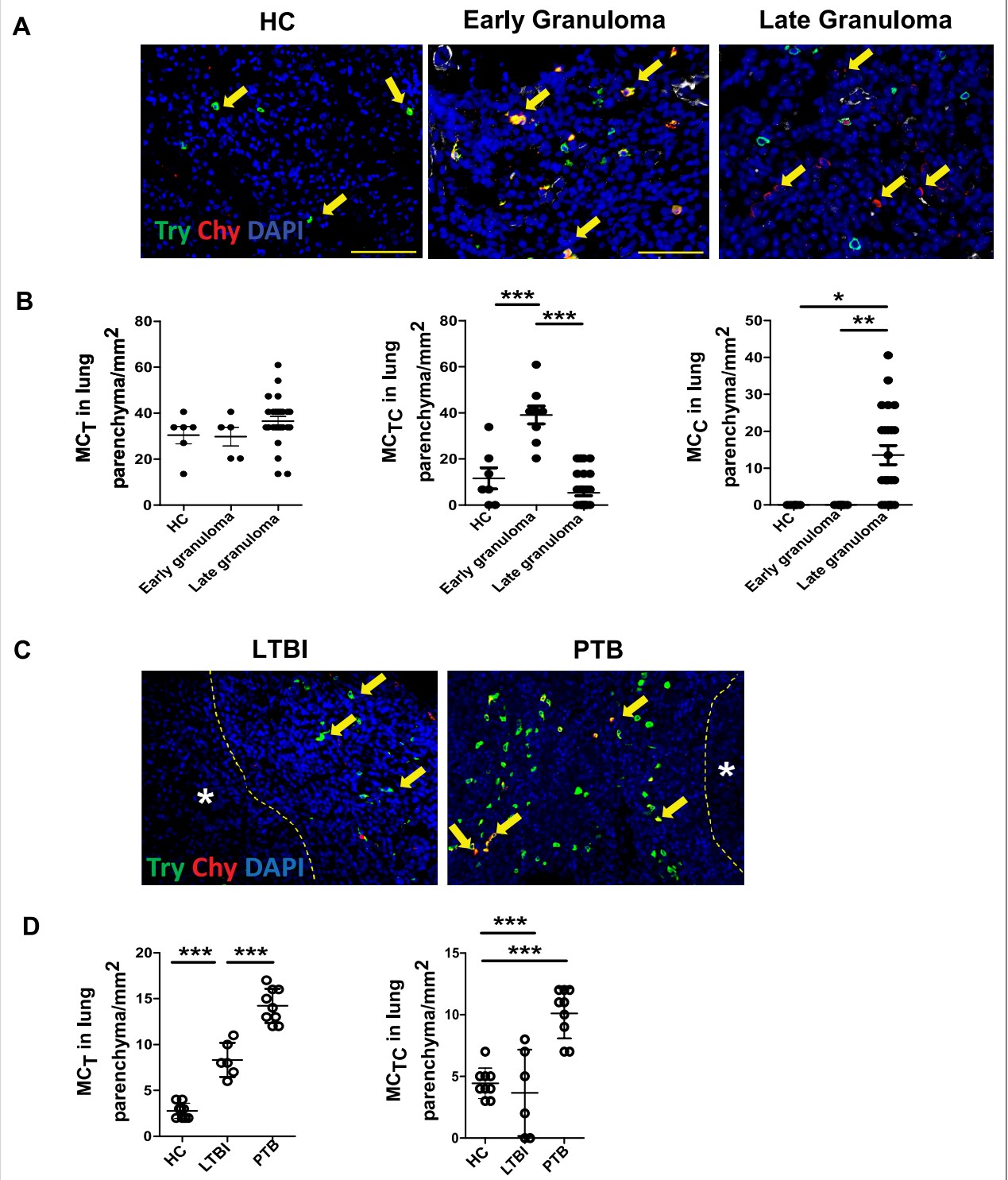

**Figure 1.** Chymase-positive mast cells (MCs) are predominant in TB-infected human and macaque lung tissue. Lung biopsies from healthy individuals (*n* = 4) or patients with PTB (*n* = 5) were stained for tryptase $MC_T$ (green) or chymase $MC_C$ (red). (**A**) Immunofluorescence microscopy shows $MC_{TS}$ (green) in healthy lung biopsies (HC). $MC_{TCS}$ (red and green merge) are located around the early granulomas, while $MC_{CS}$ (red) surround the late granulomas in TB-infected lung biopsies. (**B**) Predominance of $MC_{TS}$ in healthy lungs transitioning to $MC_{TCS}$ in early granuloma and becoming $MC_{CS}$ in late granulomas in TB-infected lungs. (**C**) Immunofluorescence microscopy shows $MC_{TS}$ (green) and $MC_{TCS}$ (merge) in lungs of healthy (HC), LTBI, and PTB macaques. (**D**) Predominance of $MC_{TS}$ (green) and $MC_{TCS}$ (merge) in PTB compared to LTBI and HC. Statistical analysis was performed using GraphPad v5, unpaired, two-tailed Student's *t*-test, ***p < 0.0001, **p < 0.001, *p < 0.05.

*Figure 1 continued on next page*

*Figure 1 continued*

The online version of this article includes the following figure supplement(s) for figure 1:

**Figure supplement 1.** Predominance of MC$_T$s in human and NHP lung interstitium, blood vessels, and bronchi.

to the more widely expressed *TPSAB1* (encoding α and β1 tryptase) differs in NHPs. Based on phylogenetic similarity to human α and β tryptase, the NHP ortholog is often referred to as α- or β-like (α/β) tryptase. However, their gene names differ, as they are still classified as putative proteins and not formally annotated with the same nomenclature as in humans. The putative tryptase genes in NHPs are annotated as *LOC699599* for *Macaca (M). mulatta* and *LOC102139613* for *M. fascicularis*. Examining these genes in the NHP single-cell transcriptome dataset, we detected the expression of the γ and the putative α/β tryptase genes, but found no expression for δ tryptase. *TPSG1* was found to be expressed at low levels and only in a few MCs from the PTB group. In contrast, the putative α/β tryptase gene was expressed in MCs across all groups, with the highest expression in PTB (*Figure 2H*), consistent with our immunofluorescence data (*Figure 1D*). As expected, the chymase (encoded by *CMA1*) expressing cells were detected exclusively in the PTB group and were absent in LTBI and HCs (*Figure 2H*). To confirm whether these *CMA1*-positive cells also co-expressed the putative α/β tryptase as well, we quantified cells expressing single and dual transcripts. This analysis revealed that most of the chymase-positive cells (243 cells) also expressed the putative α/β tryptase gene (183 cells), supporting our earlier observation of a dual tryptase–chymase MC signature linked with PTB. In contrast, MCs from HC and LTBI groups showed expression of tryptase alone (*Figure 2I*).

To strengthen our findings, we validated MCs using an independent lung single-cell transcriptome from NHP (*M. fascicularis*), collected at 4 weeks (higher bacterial load, more severe disease) and 10 weeks (low bacterial load, less severe disease) following low-dose *Mtb* Erdman infection (*Gideon et al., 2022*; *Grimbaldeston et al., 2005*). This dataset had 372 MCs at 4 weeks and 7306 MCs at 10 weeks (*Figure 2J*). We examined the expression of chymase and several tryptase genes, including *TPSG1*, *TPSD1*, and *LOC102140229* (putative NHP ortholog for human *TPSAB1*). While *TPSD1* expression was undetectable, the other tryptase genes showed high expression with a pronounced increase in chymase (*CMA1*) expression both at 4 and 10 weeks (*Figure 2K*). Similar to our transcriptomic scRNA seq dataset, we quantified MCs co-expressing *LOC102140229* and *CMA1*. Consistent with our data, *CMA1* and *LOC102140229* (the α/β-like tryptase ortholog) expressions were associated with severe disease, being significantly enriched in MCs from the higher-burden 4-week granulomas (odds ratio [OR] = 0.27, p < 1 × 10$^{-29}$, and OR = 1.68, p < 1 × 10$^{-5}$, respectively). Importantly, when restricting analysis to *CMA1*$^+$ cells, the dual-positive *CMA1*$^+$*LOC102140229*$^+$ MC subset was proportionally more abundant in less severe, 10-week granulomas (OR = 0.51, p < 1 × 10$^{-9}$) (*Figure 2M*). This suggested that while chymase (*CMA1*) expression marked severe disease, the presence of the dual tryptase–chymase phenotype in less severe lesions supported the idea that MC transcriptional diversity emerges in association with disease modulation, further supporting our observation of MC diversity with increasing dual tryptase and chymase signature associated with disease progression. Similar to our observations in *M. mulatta*, we quantified the DEGs and carried out Ucell scores in the MC subset from *M. fascicularis*. We observed that MC from high burden TB granulomas showed higher IFNγ signaling, oxidative phosphorylation (*Figure 2N, O*), similar to higher PTB scores seen in *M. mulatta*. However, unlike *M. mulatta*, *M. fascicularis* also showed increased TNF signaling in high-burden granulomas (*Figure 2P*). These results highlight that MCs in high-burden granulomas upregulated IFNγ, TNF, and metabolic programs, coupled with chymase expression, whereas less severe granulomas were enriched for tryptase-positive MCs. Although this mirrors our findings seen in *M. mulatta*, we observed species-specific differences in how TNF signaling is distributed across disease states.

## MC-deficient mice exhibit enhanced control of *Mtb*

We next determined whether MCs are induced in response to *Mtb* infection in mice and characterized their accumulation early and later in infection. In our previous studies, we observed that innate cells such as innate lymphoid cells accumulate rapidly between days 5 and 10 post-infection, followed by neutrophils, macrophages, and monocytes between days 10 and 15, and T cells by days 21 and 30 (*Ardain et al., 2019*). We found that MCs begin to accumulate in the lungs between days 21 and 30, coinciding with the timing of *Mtb* growth (*Figure 3—figure supplement 1A and B*) and T cell recruitment. To investigate the

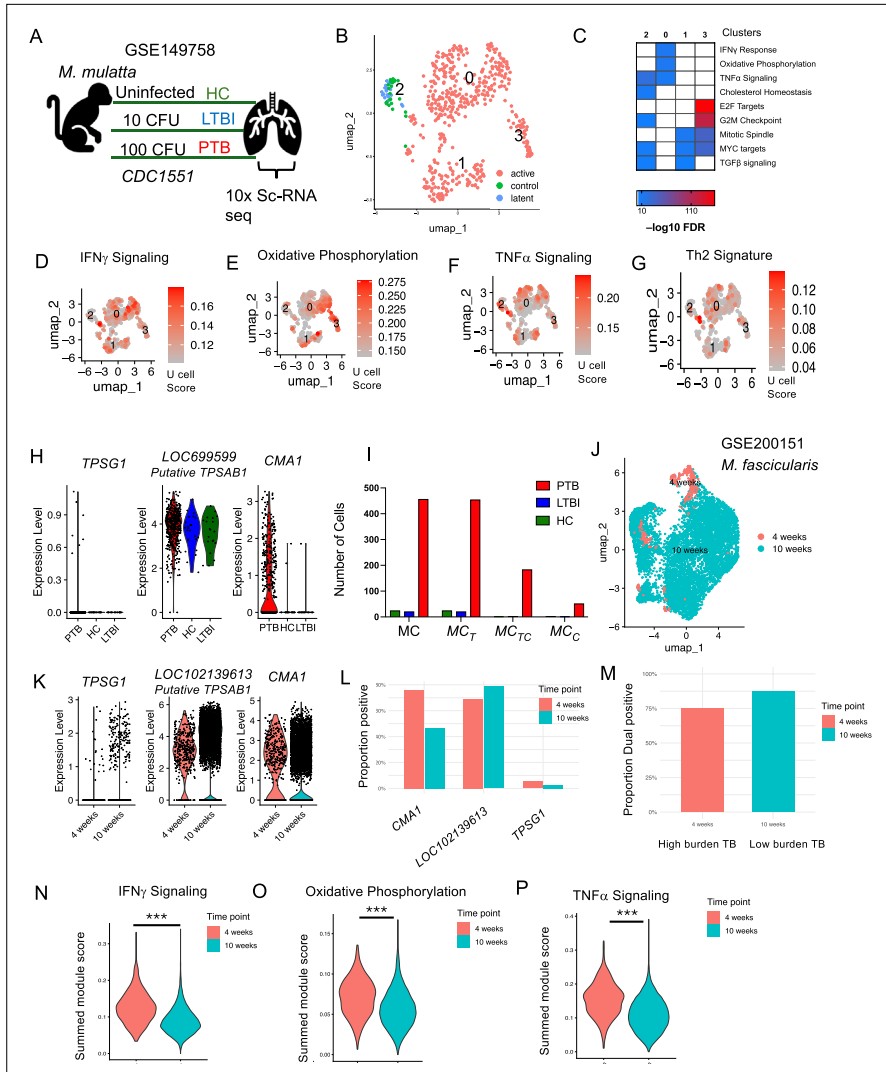

**Figure 2.** Mast cell (MC) signatures across disease conditions in NHPs. Data was reanalyzed from the lungs of *M. mulatta* infected with *Mtb CDC1551* (GSE200151). (**A**) Schematic of the study design across disease conditions (**B**) UMAP embedding of *FCER1A+* MCs, showing the distribution of these cells across the different disease conditions (PTB in pink, HC in green, and LTBI in blue). (**C**) Heatmap of Hallmark pathway analysis for differentially expressed genes, highlighting the top pathways with the highest FDR values for each condition. UCell module for pathways: IFNγ signaling (**D**), TNF-α signaling (**E**), oxidative phosphorylation (**F**), and Th2 signature (**G**) across disease conditions, shown on UMAP embeddings. (**H**) Violin plots of gene expression for key MC markers (*CMA1*, *TPSG1*, and *LOC699599*) across disease conditions. (**I**) Cell counts of different MC subtypes (MC_C, MC_T, and MC_TC) across disease conditions (PTB, red bars, LTBI, blue bars, and HC, green bars). (**J**) UMAP plot of the NHP lung granuloma dataset (GSE200151), showing the distribution of cells at 4 weeks (high disease burden) and 10 weeks (low disease burden) in *M. fasicularis* infected with *Mtb Erdman*. (**K**) Gene expression violin plots for key MC markers (*CMA1*, *TPSG1*, and *LOC699599*) from the new dataset across time points. (**L**) Proportions of different MC subtypes (MC_C, MC_T, and MC_TC). (**M**) Violin plots of summed module scores for the key pathways (IFNγ signaling, TNF-α signaling, oxidative phosphorylation) across disease burdens, showing pathway activity. Statistical significance was assessed using GraphPad v10, Kruskal–Wallis tests with Dunn's multiple comparison correction (****$p < 0.0001$, ***$p < 0.001$, **$p < 0.01$, *$p < 0.05$).

The online version of this article includes the following figure supplement(s) for figure 2:

**Figure supplement 1.** Immune cell marker expression and pathway analysis across disease conditions.

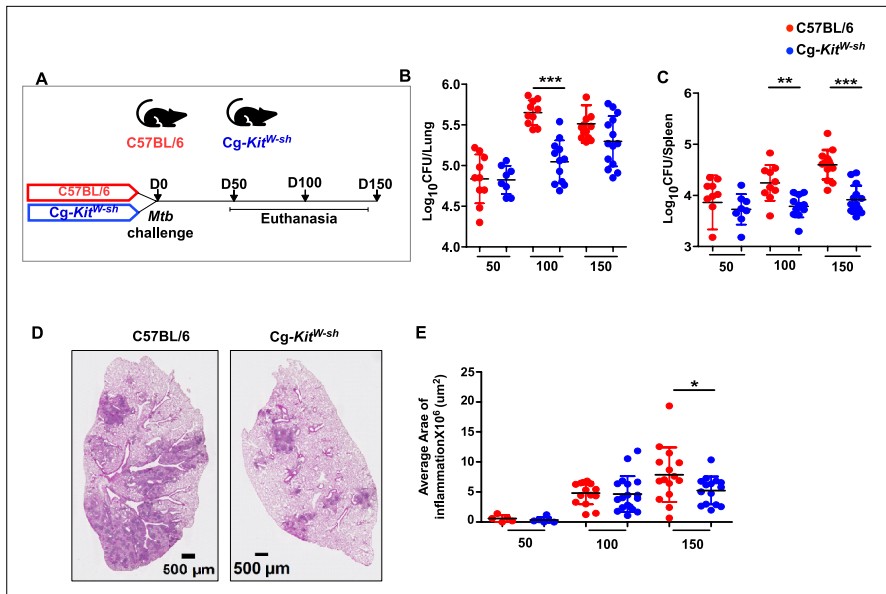

**Figure 3.** Mast cell (MC)-deficient mice are resistant to *Mtb* chronic infection. (**A**) C57BL/6 and Cg-*Kit*<sup>W-sh</sup> mice were infected with a low aerosol dose (~100 CFU) of *Mtb* HN878 and mice were sacrificed at 50, 100, and 150 dpi. (**B**) Bacterial burden was assessed in lungs and spleens by plating. (**C**) Lungs were harvested, fixed in formalin, and embedded in paraffin. Hematoxylin and eosin (H&E) staining was carried out for blinded and unbiased analysis of histopathology. (**D**) Representative images and the area of inflammation measured in each lobe are shown. Scale bars: 2 mm. Original magnification: ×20. Data points represent the mean ± SD of two experiments (*n* = 8–15 per time point per group). Statistical analysis was performed using GraphPad v5, unpaired, two-tailed Student's *t*-test between C57BL/6 and Cg-*Kit*<sup>W-sh</sup> mice, ***p < 0.0001, **p < 0.001, *p < 0.05.

The online version of this article includes the following figure supplement(s) for figure 3:

**Figure supplement 1.** Mast cells (MCs) appear at early *Mtb* infection.

functional role of MCs in *Mtb* infection, we utilized the MC-deficient mouse model, Cg-*Kit*<sup>W-sh</sup>. These mice carry spontaneous loss-of-function mutations in both alleles of the dominant *white spotting* (W) locus (i.e., s), leading to impaired c-*kit* tyrosine kinase-dependent signaling resulting in dysregulated MC development, survival, and function (*Wolters et al., 2005*). We infected Cg-*Kit*<sup>W-sh</sup> mice with low-dose aerosolized *Mtb* strain HN878 for early (50 days post-infection, dpi) and later time points (100 and 150 dpi) and compared them with C57BL/6 wild-type (WT) *Mtb*-infected mice (*Figure 3A*). While there were no significant differences observed in lung and spleen bacterial burden at 50 dpi between Cg-*Kit*<sup>W-sh</sup> and WT *Mtb*-infected mice, Cg-*Kit*<sup>W-sh</sup> *Mtb*-infected mice showed significantly lower lung and spleen *Mtb* CFU compared to WT *Mtb*-infected controls at 100 dpi. However, at 150 dpi, lung bacterial burden in Cg-*Kit*<sup>W-sh</sup> *Mtb*-infected mice trended lower but not significant; these mice showed enhanced bacterial control in the spleen (*Figure 3B, C*). The reduction in bacterial load also coincided with reduced lung inflammation in Cg-*Kit*<sup>W-sh</sup> *Mtb*-infected mice at 150 dpi (*Figure 3D, E*). These findings indicate that MC-deficient Cg-*Kit*<sup>W-sh</sup> mice exhibit improved control of *Mtb* infection during chronic stages, suggesting that MCs may contribute to disease persistence or pathogenesis in chronic TB.

## MC-deficient mice display altered innate immune cell profiles during chronic *Mtb* infection

To further address the functional basis of enhanced protection observed in MC-deficient mice, we analyzed the lung immune responses in Cg-*Kit*<sup>W-sh</sup> mice both at baselines and following *Mtb* infection, given that this mouse strain is associated with other known immune alterations (*Grimbaldeston et al., 2005*). MCs were significantly reduced in the lungs of Cg-*Kit*<sup>W-sh</sup> mice compared to WT mice at baseline (*Figure 4A*). However, we did not observe any significant differences in other innate immune populations in the lung of Cg-*Kit*<sup>W-sh</sup> mice, including DCs, recruited macrophages (RMs), alveolar macrophages (AMs), neutrophils, monocytes, and T cells at baseline compared to WT controls (*Figure 4B–F*, and *Figure 4—figure supplement 1A and B*). Following *Mtb* infection, MCs

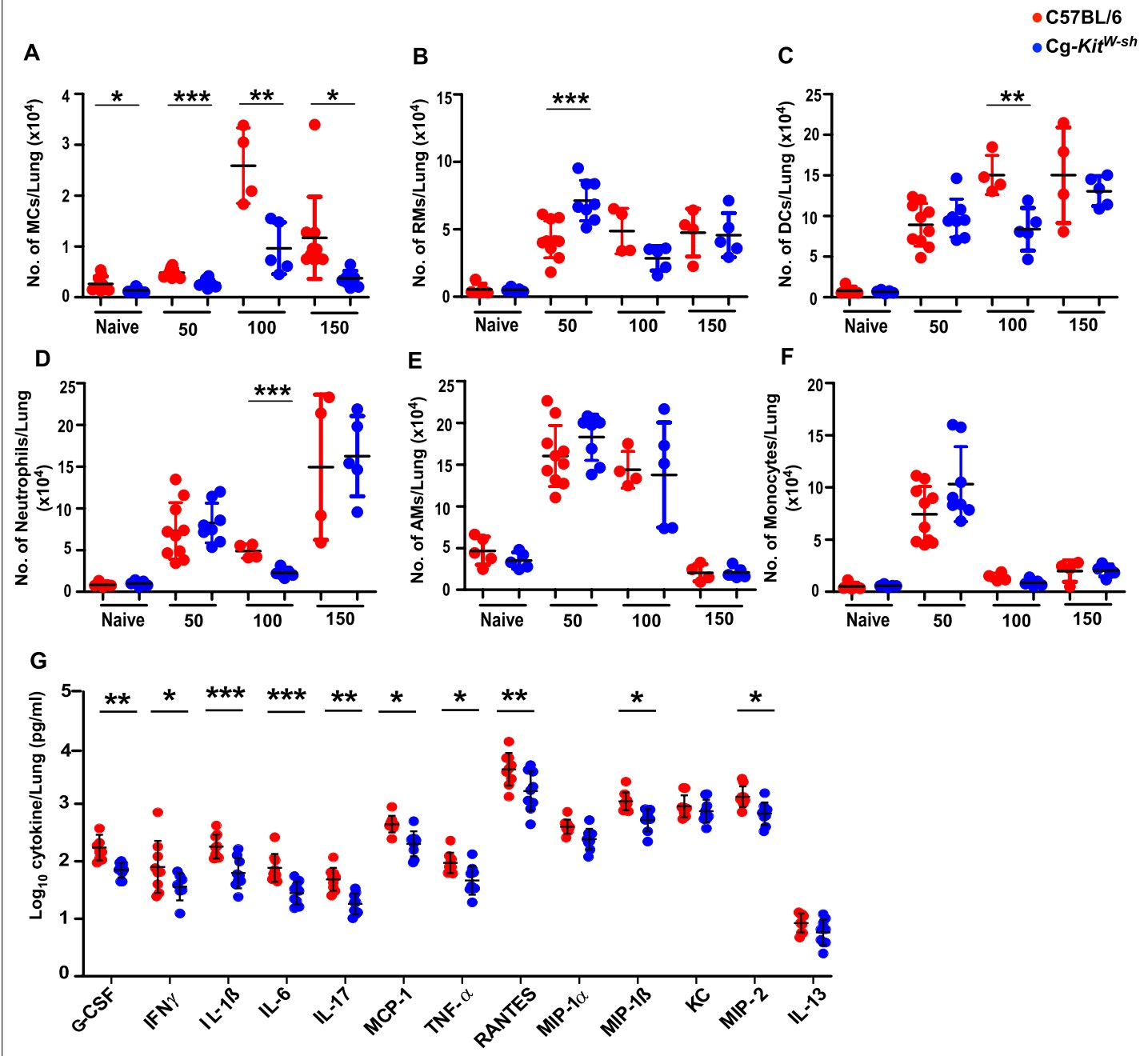

**Figure 4.** Mast cell (MC)-deficient mice have dysregulated immune profiles after *Mtb* infection. C57BL/6 and Cg-*Kit*^W-sh^ mice were infected with a low aerosol dose (~100 CFU) of *Mtb* HN878 and mice were sacrificed at 50, 100, and 150 dpi. Number of (**A**) MCs, (**B**) dendritic cells (DCs), (**C**) recruited macrophages (RMs), (**D**) neutrophils, (**E**) alveolar macrophages (AMs), and (**F**) monocytes were enumerated in the lungs of *Mtb*-infected mice. (**G**) Cytokine and chemokine production in lung homogenates from mice, collected at 150 dpi, was assessed by multiplex cytokine analysis. Data points represent the mean ± SD of 1 of 2 individual experiments (*n* = 4–10 per time point per group). Statistical analysis was performed using GraphPad v5, unpaired, two-tailed Student's *t*-test between C57BL/6 and Cg-*Kit*^W-sh^ mice, for (**A–F**), ***$p < 0.0001$, **$p < 0.001$, *$p < 0.05$; and using GraphPad v10, two-way ANOVA Sidak's multiple comparison test for (**G**) ****$p < 0.0001$, ***$p < 0.001$, **$p < 0.01$, *$p < 0.05$. Outliers were removed from the subsets using Grubb's outlier test.

The online version of this article includes the following figure supplement(s) for figure 4:

**Figure supplement 1.** Mast cell (MC)-deficient mice have no baseline differences in T cell numbers.

**Figure supplement 2.** Mast cell (MC)-deficient mice have reduced numbers of activated CD4+ and CD8+ T cells in the lung.

accumulated progressively in the lungs of both WT and Cg-*Kit*[W-sh] mice up to 100 dpi, after which their numbers stabilized through 150 dpi. In contrast, MC accumulation was significantly impaired in Cg-*Kit*[W-sh] mice throughout infection (**Figure 4A**). At 50 dpi, RMs were elevated in Cg-*Kit*[W-sh] mice, with no significant difference observed in DCs or neutrophils compared to WT mice (**Figure 4B**). By 100 dpi, both DCs and neutrophils were decreased in Cg-*Kit*[W-sh] mice; however, these changes were not sustained at 150 dpi (**Figure 4C, D**). Across all time points, AMs and monocyte populations remained comparable between Cg-*Kit*[W-sh] and WT *Mtb*-infected mice (**Figure 4E, F**). Previous studies have implicated MCs in driving T cell responses (**Elieh Ali Komi and Grauwet, 2018**); therefore, we next examined T cell responses induced post-infection. We found no differences in the activated CD4[+] and CD8[+] T cell responses at 50 dpi; however, by 100 dpi, both populations were significantly reduced at 100 dpi in Cg-*Kit*[W-sh] *Mtb*-infected mice (**Figure 4—figure supplement 2A and E**). This reduction extended to functional subsets, with fewer CD4[+] T cells producing IFNγ, as well as diminished dual TNF-α and IFNγ producing cells in the Cg-*Kit*[W-sh] *Mtb*-infected mice as compared to WT *Mtb*-infected mice (**Figure 4—figure supplement 2B and D**). We did not find any significant differences in the CD8[+] T cells producing IFNγ, TNF-α, and dual IFNγ and TNF-α producing cells in the Cg-*Kit*[W-sh] *Mtb*-infected mice (**Figure 4—figure supplement 2F-H**). Finally, we measured cytokine responses in the lungs of Cg-*Kit*[W-sh] and WT *Mtb*-infected mice at 150 dpi. We found that proinflammatory cytokines that direct monocyte/macrophage and T cell responses (G-CSF, IFNγ, IL-1β, IL-6, IL-17, MCP-1, TNF-α, and RANTES) were significantly lower in Cg-*Kit*[W-sh] *Mtb*-infected lungs compared to WT *Mtb*-infected lungs. Similarly, chemokines driving neutrophil recruitment (MIP-1β and MIP-2) were reduced in Cg-*Kit*[W-sh] *Mtb*-infected mice, whereas Th2 cytokines such as IL-13 did not differ between the two groups (**Figure 4G**). Together, these results provide evidence that MCs are induced following *Mtb* infection, accumulate in the lung, and mediate immune responses that drive pathology and promote *Mtb* susceptibility and dissemination during chronic TB.

## Transfer of MCs into lung airways promotes neutrophil accumulation and *Mtb* dissemination

To examine whether MCs can impact *Mtb* control and dissemination, we adoptively transferred bone marrow-derived MCs into the airways of WT mice, hereafter referred to as B6[MC] mice. A total of 5 × 10⁴ MCs were transferred, approximating the number of MCs present in the lungs of WT *Mtb*-infected mice at 100 dpi. Following transfer, mice were infected with *Mtb*, and we found that MCs were retained in the lungs of B6[MC] *Mtb*-infected mice up to 30 dpi (**Figure 5A**). Strikingly, B6[MC] *Mtb*-infected mice showed increased neutrophil frequencies and reduced RMs when compared with B6 *Mtb*-infected mice (**Figure 5B, C**). However, no differences in the total numbers of lung MCs, neutrophils, RMs, or DCs were observed in B6 and B6[MC] groups (**Figure 5—figure supplement 1**). While the presence of additional MCs in the lungs did not alter the bacterial burden (*Mtb* CFU) and inflammation in the lungs, B6[MC] *Mtb*-infected mice showed a significant increase in the *Mtb* CFUs in the spleen, suggesting a role for MCs in promoting dissemination from the lungs (**Figure 5D–F**). Consistently, histological analysis revealed greater neutrophil infiltration in lung sections of B6[MC] *Mtb*-infected mice compared to WT *Mtb*-infected mice (**Figure 5G, H**), implicating MCs in driving neutrophil recruitment and potentially enhancing the systemic spread of *Mtb*.

## Discussion

The immune mechanism(s) that mediate the progression from LTBI to PTB are unclear. In this study, we identified MCs as an innate cell type that is overrepresented during PTB, transcriptionally expressed signatures associated with IFNγ, oxidative phosphorylation, and MYC signaling, and localized within mature TB granulomas. Importantly, using mice deficient in MCs, we showed a potential pathological role for MCs in mediating susceptibility to TB, thus providing MCs as a novel therapeutic target.

MCs have been shown to interact with *Mtb* through the GPI-anchored molecule CD48 (**Muñoz et al., 2003**), interaction with TLR2 (**Carlos et al., 2007**), and potentially TLR4 (**McCurdy et al., 2001**). Additionally, *Mtb* is also thought to be internalized by lipid rafts on MCs (**Muñoz et al., 2009**), thus serving as a long-lasting reservoir for *Mtb* (**da Silva et al., 2014**). These in vitro studies have shown that MC exposure to *Mtb* results in degranulation of MCs, as well as the induction of proinflammatory cytokines such as TNF-α and IL-1β. Consistently, another study using the same route of high-dose

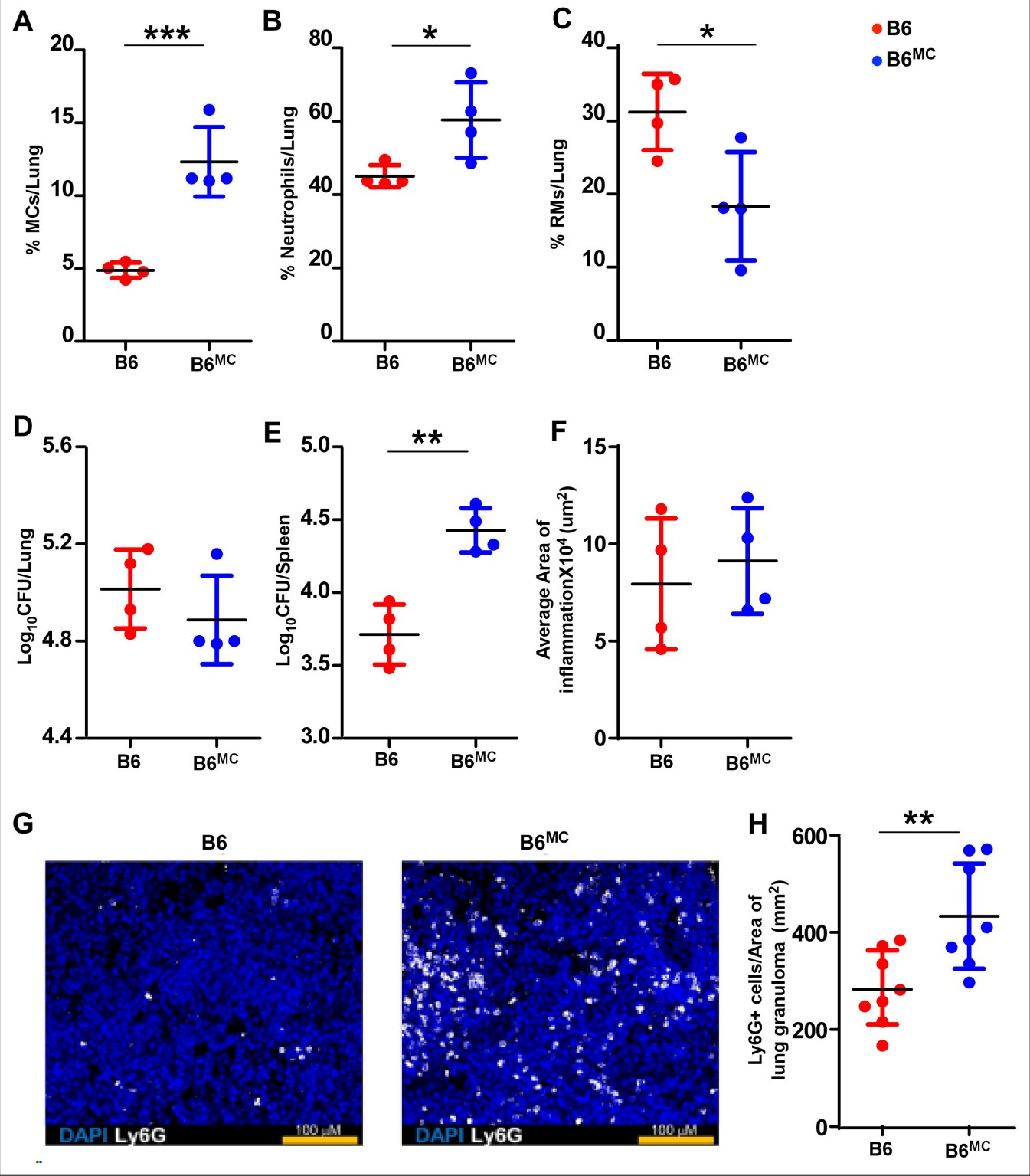

**Figure 5.** Wild-type mice with airway transferred mast cells (MCs) promote bacterial dissemination. Bone marrow-derived in vitro cultured MCs (5 × 10^4 cells/mouse) were adoptively transferred into the lung airways of C57BL/6 mice 1 day before infecting with a low aerosol dose (~100 CFU) of *Mtb* HN878. MCs were replenished in these mice at 21 dpi, and mice were sacrificed at 30 dpi. Frequencies of (**A**) MCs, (**B**) neutrophils, and (**C**) recruited macrophages (RMs) were enumerated in the lungs of *Mtb*-infected mice. Bacterial burden was assessed in (**D**) lungs and (**E**) spleens by plating. (**F**) Lungs were harvested, fixed in formalin, and embedded in paraffin. Hematoxylin and eosin (H&E) staining was carried out for blinded and unbiased analysis of histopathology. (**G**) Immunofluorescence microscopy shows more neutrophil infiltration in the lungs of MC-transferred WT mice. (**H**) Ly6G+ cells per

*Figure 5 continued on next page*

*Figure 5 continued*

area of lung granuloma measured in each lobe are shown. Scale bars: 2 mm. Original magnification: ×20. Data points represent the mean ± SD, of 1 of 2 individual experiments (n = 4 per group). Statistical analysis was performed using GraphPad v5, an unpaired, two-tailed Student's *t*-test between the groups, ***p < 0.0001, **p < 0.001, *p < 0.05.

The online version of this article includes the following figure supplement(s) for figure 5:

**Figure supplement 1.** Lung myeloid cell accumulation did not vary in mast cell (MC)-transferred WT *Mtb*-infected mice.

infection reported increased bacterial burden in both the lungs and spleen of MC-deficient Cg-*Kit*[W-sh] mice compared to wild-type C57BL/6 controls (*Villareal-Rivota et al., 2025*). While these studies using a high-dose model of infection reported early induction of inflammatory mediators from MC within hours to days, our in vivo results using a physiological low dose of *Mtb* infection model showed that MCs accumulated between 21 and 30 days, coinciding with the onset of T cells in the lung. Interestingly, despite the accumulation of MCs in the lung at 30 dpi following low-dose aerosol infection, the impact of MC deficiency on *Mtb* control and inflammation in Cg-*Kit*[W-sh] mice was not evident until 100 dpi. This is similar to our published studies where we found that S100A8/9 deficiency resulted in reduced neutrophil lung accumulation (*Scott et al., 2020*), resulting in improved *Mtb* control and improved TB disease, but after 100 dpi. However, this is in contrast to the role of eosinophils in TB, as eosinophil deficiency resulted in increased *Mtb* CFU (*Bohrer et al., 2021*). MCs were the primary innate cell type that was defective in the lungs of Cg-*Kit*[W-sh] mice at baseline and throughout infection, and their absence was associated with better containment of *Mtb* in both lung and spleen, indicating a pathological role for MCs during chronic TB.

Our data also showed that MCs accumulated in the lungs of C57BL/6 mice from 50 to 100 dpi, after which their numbers stabilized. This increased accumulation of MCs during chronic infection coincided with elevated infiltration of DCs and neutrophils. Importantly, we showed that enhancing MC numbers through adoptive transfer in the lungs of WT *Mtb*-infected mice similarly promoted neutrophil accumulation and facilitated *Mtb* dissemination. This suggests that MCs are not merely a consequence of chronic inflammation, but active modulators capable of shaping immune cell dynamics early in infection. When present in sufficient numbers, MCs with their ability to promote neutrophil recruitment can potentially influence the balance of protective versus pathological inflammation during TB. These results raise the possibility that MC accumulation during chronic infection serves to fine-tune the composition of the innate immune compartment in the lung. Whether this regulation is beneficial or detrimental to host control of *Mtb* remains to be explored. However, our data suggested that manipulating MC responses may offer a novel avenue for modulating immune dynamics in TB, particularly in the chronic phase, where inflammation must be tightly regulated to prevent tissue damage. Additionally, the reduced inflammation observed at 150 dpi is associated with decreased levels of proinflammatory cytokines and chemokines that recruit monocytes/macrophages, T cells, and neutrophils. Overall, based on our results along with the current literature, we propose pathological roles for neutrophils and MCs, while other granulocytes, such as eosinophils, may mediate protective roles (*Bohrer et al., 2021*).

MCs can release cytokines and chemokines, antimicrobial peptides, and granules upon pathogen sensing and to control pathogens (*Naqvi et al., 2017*; *Naqvi et al., 2021*; *Naqvi et al., 2020*). In the context of *Mtb* exposure, MCs have been shown to undergo degranulation, including histamine and β-hexosaminidase (*Muñoz et al., 2003*). Indeed, histamine-deficient mice showed decreased neutrophils, as well as proinflammatory cytokine production following *Mtb* infection (*Carlos et al., 2009*). Additionally, induction of degranulation following intratracheal *Mtb* infection resulted in reduced proinflammatory cytokines as well as reduced lung inflammation. Similarly, our study showed reduced DCs, neutrophils, and CD4[+] and CD8[+] T cell responses in the lungs of MC-deficient *Mtb*-infected mice, along with reduced proinflammatory cytokines and lung inflammation at chronic time points. Thus, together with published work, MCs can potentially modulate neutrophils and other inflammatory mediators in high-dose models, as well as in a physiologically relevant *Mtb* infection model, leading to disease pathology. The exact mechanism by which MCs contribute to the pathology, dissemination, and promotion of *Mtb* infection is an area of future investigation.

At baseline, human lungs have been reported to primarily express tryptase (*Poto et al., 2022*). Indeed, we found that this is true for both macaque and human lungs in our study, where healthy lungs expressed MC[T]s. Additionally, we found that early granulomas and in LTBI, we saw expression of

MC$_{TC}$s with a switch to more and accumulation of MC$_C$s in late-stage granulomas. Chymase expression may modulate extracellular matrix components (ECM) such as fibronectin, leading to tissue remodeling, impacting cellular communication, and inducing cleavage for key cytokines such as IL-6, IL-13, IL-15, and IL-33, as well as TGF-β. (*Pejler, 2020*; *Waern et al., 2013*). Studies have shown that tryptase can induce proliferation of fibroblasts, epithelial cells, and smooth muscle cells, causing airway remodeling during diseased conditions (*Mogren et al., 2021*). Tryptase can also inactivate a large range of peptides by cleaving specific substrates, such as fibrinogen, gelatin, pro-matrix metalloproteinases, and complement factors, thus moderating inflammatory responses (*Caughey, 2007*). Based on our results from human and macaque lung, we hypothesize that MC$_{TC}$s may synergize to drive responses induced by both pathways at early time points and possibly just by MC$_C$s at later time points. Additional studies describing these subsets and testing their functional relevance in in vivo models are future steps in delineating the role of these subsets in TB.

Single-cell transcriptomic analysis revealed a differential activation state between the MCs that accumulate in lungs in PTB as compared to LTBI and HC in NHPs (*Esaulova et al., 2021*). The MCs from PTB animals showed a closer resemblance to MC$_C$s, characterized by higher IFNγ, metabolic activation, and chymase signatures, confirming that chymase expression is associated with disease severity. This association of chymase expression with disease severity was confirmed in an independent single-cell lung dataset (*Gideon et al., 2022*), where the analysis revealed similar enrichment of chymase-expressing MC$_C$s in granulomas with higher disease burden, accompanied by similar activation of IFN-γ and metabolic pathways. Metabolic activation of oxidative phosphorylation, as observed here, has been associated with activated MCs, and inhibiting mitochondrial ATP production reduced MC degranulation and cytokine production (*Paruchuru et al., 2022*; *Sharkia et al., 2017*). This also suggests that MCs, which accumulate in the lungs in LTBI and HC, were not only lower in proportion but also less activated, meaning that TB-induced activation of MCs contributes to their pathogenic phenotype in disease. Our data aligned with previous observations by *Garcia-Rodriguez et al., 2021*, showing that chymase-expressing MCs accumulate in TB-induced lung lesions and may contribute to fibrotic processes surrounding granulomas (*Garcia-Rodriguez et al., 2021*). We also observed an expansion of MC$_{TC}$s in the PTB group in NHPs, mirroring the phenotypic shift from M$_{CT}$ to MC$_{TC}$ seen in human lung sections. While Garcia-Rodriguez et al. suggested this shift occurs in fibrotic areas and may reflect tissue remodeling, they did not demonstrate improved lung function. Although fibrosis was not directly measured in our study, these findings support the association of chymase-positive MCs with advanced, inflammatory disease, reinforcing their potential role in TB pathogenesis.

MCs are known to gear toward a Th2 signature with increased chymase expression (*Toru et al., 1998*). This increase was reflected in MCs from the macaque lung, showing a high transcriptomic Th2 signature in PTB but not in clusters found in LTBI and HC. In essence, transcriptomics reflected the hyperactivated nature of the MCs in PTB, which might make them more pathological during infection. Although our in vivo mouse study showed a pathological role of MCs, significant secretion of Th2 cytokines was not seen, likely because the C57BL/6 mouse model is prone toward Th1 polarization (*Wakeham et al., 2000*). MCs can release both preformed and de novo synthesized TNF-α, hence helping in early bacterial clearance (*Gordon and Galli, 1991*). Similarly, our study showed that the MCs from HC and LTBI individuals expressed higher levels of TNF-α and were in a less metabolically activated state (lower OXPHOS signature). So, additional studies will help in elucidating the mechanism through which MCs mediate pathological roles.

In summary, we demonstrated that the accumulation of chymase-producing MCs in PTB is a cross-species phenomenon that contributes to increased TB pathology and loss of TB control, thereby elucidating the pathological role of MCs in the control of *Mtb* infection. By targeting MC pathways or signaling mechanisms, host-directed therapies hold the promise of enhancing the effectiveness of existing treatments and mitigating disease-related complications.

## Materials and methods
### Study subjects and animal studies
All human lung biopsy samples were obtained from the Tuberculosis Outpatient Clinic and the Department of Pathology at the National Institute of Respiratory Diseases (INER) in Mexico City, before *Mtb* treatment with informed consent, and with the approved protocol by the INER IRB for their use

(project numbers B04-15 and B09-23). Also, lung samples from healthy controls (HC), non-TB individuals, were obtained from the tissue repository of the Department of Pathology at INER. No compensation was provided to the patients. The analysis was conducted at Washington University in St. Louis, School of Medicine, and approved by the University IRB (reference number 201811050).

Non-human primate procedures were approved by the Institutional Animal Care and Use Committee of Tulane National Primate Research Center and were performed following National Institutes of Health (NIH) guidelines. Male and female Indian rhesus macaques, aged 4–16 years, verified to be free of *Mtb* infection by tuberculin skin test, were obtained from the Tulane National Primate Research Center. The animals were housed in an ABSL3 facility.

C57BL/6 and B6.Cg-*Kit^W-sh*/HNihrJaeBsmJ mice (Strain #:030764) were procured from Jackson Laboratory (Bar Harbor, ME) and bred at Washington University in St. Louis or University of Chicago. Six- to eight-week-old female and male mice were used in the experiments. All mice were maintained and used per the approved Institutional Animal Care and Use Committee (IACUC) guidelines at Washington University in St. Louis (IACUC approval 20190101) or University of Chicago (IACUC approval 72713).

## Aerosol infection

For murine experiments, *Mtb* strain HN878 (Source: BEI Resources) was cultured in Proskauer Beck medium containing 0.05% Tween 80 until reaching mid-log phase and frozen in 1 ml aliquots at –80°C until used. Mice were aerosol infected with ~100 colony-forming units (CFU), as described previously (*Khader et al., 2007*). *Mtb* strain CDC1551 (Source: BEI Resources) was used to infect NHPs. This species-specific choice reflects differences in pathogenicity: HN878 induces robust disease in mice, while CDC1551, a less virulent strain, allows development of a macaque model that recapitulates latent and chronic TB upon low- to moderate-dose aerosol exposure, respectively (*Kaushal et al., 2015*; *Sharan et al., 2021*; *Singh et al., 2025*). This ensures physiologically relevant and controlled studies within each species. Macaques were assigned to three groups: (1) uninfected control, (2) macaques with LTBI were exposed to a low dose (~10 CFU), and (3) macaques with PTB were exposed to a high dose (~100 CFU) of *Mtb* CDC1551 via the aerosol route using a custom head-only dynamic inhalation system housed within a class III biological safety cabinet as previously described (*Esaulova et al., 2021*). The animals were periodically monitored for their physiological parameters and to monitor disease symptoms.

## Bacterial burden and cytokine analysis

Bacterial burden was assessed using serial 10-fold dilutions of lung or spleen homogenates and plated on 7H11 agar solid medium supplemented with OADC (oleic acid, bovine albumin, dextrose, and catalase). Colonies were counted after 2–3 weeks of incubation. Cytokine/chemokine expression was analyzed in lung homogenates from infected mice via Luminex (Millipore-Sigma) or ELISA (R&D) as per the manufacturer's protocol.

## Generation of single-cell suspensions from tissues and flow cytometry staining

Lung single-cell suspensions from *Mtb*-infected mice were prepared as previously described (*Gopal et al., 2013*). Briefly, mice were euthanized with $CO_2$. The right lower lobe was isolated and perfused with heparin in saline. Lungs were minced and incubated with collagenase/DNase for 30 min at 37°C. Lung tissue was pushed through a 70-µm nylon screen to obtain a single-cell suspension. Following lysis of erythrocytes, the cells were washed and resuspended in cDMEM (DMEM high glucose + 10% fetal bovine serum + 1% penicillin/streptomycin) for flow cytometry staining. For flow cytometric analysis, cells were either stained immediately or stimulated with phorbol myristate acetate (PMA-50 ng/ml; Sigma-Aldrich) and ionomycin (750 ng/ml; Sigma-Aldrich) in the presence of GolgiStop (BD Pharmingen).

For myeloid cell surface staining, the following fluorochrome-conjugated antibodies were used: CD11b-APC (clone M1/70), CD11c-PE-Cy7 (clone HL3, BD Biosciences), GR-1-PerCP-Cy5.5 (clone RB6-8C5, BD Pharmingen), and MHC class II-FITC (clone M5/114.15.2, Tonbo Biosciences), CD117 (cKit)-Super Bright 780 (clone 2BB, eBioscience), and FcεR1-PE (clone MAR-1, eBioscience). Myeloid cell subsets were defined as follows: AMs (CD11c⁺CD11b⁻), neutrophils (CD11b⁺CD11c⁻Gr-1^hi),

monocytes (CD11b$^+$CD11c$^-$Gr-1$^{med}$), RMs (CD11b$^+$CD11c$^-$Gr-1$^{low}$), and MCs (CD11b$^-$cKit$^+$FcεR1$^+$). T cells were identified based on a gating strategy as described before (*Griffiths et al., 2016*). Surface staining included CD3-AF700 (clone 500A2, BD Biosciences), CD4-Pacific Blue (clone RM4.5, BD Biosciences), CD44-PE-Cy7 (clone 1M7, Tonbo Biosciences), and CD8-APC-Cy7 (clone 53–6.7, BD Biosciences). For intracellular cytokine staining, lung cells were fixed and permeabilized using fixation/permeabilization concentrate and diluent (eBioscience) following the manufacturer's instructions. Cells were then stained with IFNγ-APC (clone XMG1.2, Tonbo Biosciences) and TNF-α-FITC (clone MP6-XT22, BD Pharmingen), or respective isotype controls (APC rat IgG1κ and FITC rat IgG1α, BD Pharmingen) for 30 min. Samples were acquired on a four-laser BD Fortessa Flow Cytometer, and data were analyzed using FlowJo software (Treestar). Absolute cell numbers for each population were back-calculated based on total viable cell counts per lung sample.

## In vitro culture and intratracheal delivery of MCs

Six- to eight-week-old C57BL/6 mice were euthanized by $CO_2$, and femurs and tibia were collected. Bone marrow cells obtained after RBC lysis were suspended in bone marrow-derived MC (BMMC) media containing RBMI media (Gibco) supplemented with 20% FBS (Sigma), 4 mM glutamine (Sigma), 25 mM HEPES (Corning), $5 \times 10^{-5}$ 2-mercaptoethanol (Sigma), 1 mM sodium pyruvate (Sigma), 0.1 mM nonessential amino acids (Gibco), penicillin and streptomycin (Sigma), 20 ng/ml murine IL-3 and 20 ng/ml murine Stem Cell factor (Peprotech) and maintained in BMMC media at $1 \times 10^6$ cells per ml in T75 flasks at 37°C in 5% $CO_2$ incubator. Cells were fed twice every week with fresh BMMC media by centrifuging non-adherent cells at 1000 rpm for 10 min at room temperature, resuspending at a density of $1 \times 10^6$/ml, and were maintained for 30 days. At 30 days, more than 95% cells were positive for FcεRI and c-kit, which was confirmed by flow cytometry (*Varma and Puri, 2019*). $5 \times 10^4$ BMMC having 95% FcεRI and c-kit positivity were transferred in C57BL/6 through the intratracheal route.

## Morphometric analysis of lung histopathology and neutrophil infiltration

For mouse studies, the left upper lobe was collected for histomorphometric analysis. The lobes were infused with 10% neutral buffered formalin and embedded in paraffin. 5-µm-thick lung sections were cut using a microtome, stained with hematoxylin and eosin, and processed for light microscopy. Images were captured using the automated Nanozoomer digital whole slide imaging system (Hamamatsu Photonics). Regions of inflammatory cell infiltration were delineated utilizing the NDP view2 software (Hamamatsu Photonics), and the percentage of inflammation was calculated by dividing the inflammatory area by the total area of individual lung lobes. All scoring was conducted in a blinded manner. Formalin-fixed paraffin embedded (FFPE) lung sections were also stained with APC-conjugated rat anti-mouse Ly6G (clone 1A8, BioLegend, RRID:AB_2227348). Nuclei were counterstained with DAPI. Neutrophils were quantified in three randomly selected 200× fields per lung section. Images at ×200 magnification were acquired using a Zeiss Axioplan microscope and recorded with a Hamamatsu camera.

FFPE lung sections from healthy individuals, TB patients, and NHP infected with *Mtb* were stained with goat anti-human MC chymase (LifeSpan Biosciences, LS-B4134, RRID:AB_10718418) and rabbit anti-human tryptase (Cell Signaling Technology, 195235). Primary antibodies were detected with Alexa Fluor 568 donkey anti-goat IgG (Thermo Fisher Scientific, A-11057, RRID:AB_2534104) and Alexa Fluor 488 donkey anti-rabbit IgG (Jackson ImmunoResearch Laboratories, 711-546-152, RRID:AB_2340619). Nuclei were labeled with DAPI. MC positive for chymase, tryptase, or both were blindly quantified in three 200x random fields per sample in human and NHP lung sections. 200x pictures were taken with an Axioplan Zeiss microscope and recorded with a Hamamatsu camera.

## Single-cell data reanalysis

The NHP single-cell lung data was accessed from GEO (GSE149758) and processed through Cell Ranger v7.0 using the *Macaca mulatta* reference genome (Mmul_10). The obtained matrix file was processed through the R package *Seurat v5* for downstream analysis of the count matrix. The cells were filtered based on mitochondrial gene content and were selected for analysis when at least 500 genes were detected. Data was log normalized. The most variable genes were detected by the *FindVariableFeatures* function and used for subsequent analysis. Latent variables (number of UMIs and

mitochondrial content) were regressed out using a negative binomial model (function *ScaleData*). Principal component analysis (PCA) was performed with the *RunPCA* function. A UMAP dimensionality reduction was performed on the scaled matrix (with most variable genes only) using the first 20 PCA components to obtain a two-dimensional representation of the cell states. For clustering, we used the functions *FindNeighbors* (20 PCA) and *FindClusters* (resolution 0.25). MCs were identified and re-clustered based on expression of the canonical MC marker genes *FCER1A* (High affinity Fc IgE receptor), *CD48*, *FCER1G* (Fc IgE receptor), *MS4A2* (IgE subunit), and *ITGAX* (CD11c) as a negative marker. The cells identified as the MC cluster (only one cluster) were subset and re-clustered using the method outlined above at a resolution of 0.1. To identify marker genes for MCs, we used *FindAllMarkers* to compare the cluster against all other clusters and *FindMarkers* to compare selected clusters. For each cluster, only genes that were expressed in more than 15% of cells with at least 0.15-fold differences were considered. The differential genes were subjected to enrichment analysis using the Hallmark pathway gene set from *MsigDB*. Only the pathways that met an FDR threshold less than 0.05 were considered. Gene signatures were defined with the R package *Ucell*. The output is a module signature score generated by the *AddModuleScore* function. The obtained score was overlaid on the UMAP and visualized. The values per cell were extracted and used to plot a summed module U cell score. GraphPad Prism was used for the violin plots and the heatmap. All other figures were generated in R.

An independent *M. fascicularis* lung single-cell RNA-seq dataset (GSE200151) (*Gideon et al., 2022*) was downloaded and processed using Cell Ranger v7.0 with the *M. mulatta* reference genome (Mmul_10). The resulting count matrices were imported into Seurat v5 for downstream analysis. Cells were filtered based on mitochondrial gene content and were retained if they expressed at least 500 genes. Data were log-normalized, the most variable genes were identified with *FindVariableFeatures*, and confounding effects of sequencing depth (UMI counts) and mitochondrial fraction were regressed out during scaling with ScaleData. PCA was performed, followed by clustering (*FindNeighbors, FindClusters*) and UMAP embedding (*RunUMAP*). MCs were identified and subset based on canonical marker expression (*FCER1A*, *CD48*, *FCER1G*, and *MS4A2*) and re-clustered at low resolution to ensure purity of the MC population. To quantify MC subsets across disease severity states, we calculated the proportion of cells expressing chymase (*CMA1*), tryptase (*LOC102140229*, *TPSG1*), or dual-positive $CMA1^+LOC102140229^+$ using binary thresholds (>0 counts). Fisher's exact tests were performed to compare proportions between MCs from high-burden granulomas (4 weeks, more severe disease; $n = 372$ MCs) and low-burden granulomas (10 weeks, less severe disease; $n = 7306$ MCs). For each comparison, we reported the OR with 95% confidence interval and the exact p-value. To examine functional programs, we computed UCell signature scores for Hallmark IFNγ signaling, TNF signaling, and oxidative phosphorylation. Signature score distributions were compared between severity groups. All statistical tests were performed in R (v4.3.1) with the Seurat (v5.0) and UCell (v2.0) packages. Violin and bar plots were generated in R or exported to GraphPad Prism for visualization.

## Data analysis and statistics

All data were analyzed using the indicated methodology in each figure legend. A two-sided unpaired *t*-test was performed for comparing the significance between two groups, one-way ANOVA Tukey's test, and Sidak's multiple comparison test were performed for more than two groups using GraphPad Prism 5 and 10, respectively (La Jolla, CA). Significance is denoted on the figure and the respective figure legends. Outliers, if any, were removed using Grubb's outlier test and mentioned in the respective figures.

## Acknowledgements

This work was supported by Washington University in St. Louis; NIH grants HL105427, AI111914, AI134236, and AI123780 to SAK, and the Department of Microbiology, University of Chicago. We thank Ms. Lan Lu, Washington University in St. Louis, and Adlai Politi, University of Chicago, for technical support and assistance.

## Additional information

### Funding

| Funder | Grant reference number | Author |
| --- | --- | --- |
| National Institutes of Health | HL105427 | Shabaana A Khader |
| National Institutes of Health | AI111914 | Shabaana A Khader |
| National Institutes of Health | AI134236 | Shabaana A Khader |
| National Institutes of Health | AI123780 | Shabaana A Khader |

The funders had no role in study design, data collection, and interpretation, or the decision to submit the work for publication.

### Author contributions

Ananya Gupta, Conceptualization, Data curation, Formal analysis, Methodology, Writing – original draft; Vibha Taneja, Data curation, Formal analysis, Methodology, Writing – original draft, Writing – review and editing; Javier Rangel-Moreno, Yun Tao, Kuldeep Singh Chauhan, Data curation; Nilofer Naqvi, Methodology, Writing – review and editing;  Abhimanyu, Data curation, Methodology, Writing – review and editing; Mushtaq Ahmed, Data curation, Supervision, Methodology; Daniela Trejo-Ponce de Leon, Gustavo Ramírez-Martínez, Luis Jiménez-Alvarez, Cesar Luna-Rivero, Resources; Joaquin Zuniga, Resources, Data curation, Methodology; Deepak Kaushal, Resources, Data curation, Supervision, Methodology, Writing – review and editing; Shabaana A Khader, Conceptualization, Resources, Data curation, Software, Formal analysis, Supervision, Funding acquisition, Investigation, Visualization, Methodology, Writing – original draft, Project administration, Writing – review and editing

### Author ORCIDs

Ananya Gupta http://orcid.org/0000-0001-7917-7877
Vibha Taneja http://orcid.org/0009-0002-9849-360X
Javier Rangel-Moreno http://orcid.org/0000-0002-9738-1182
Nilofer Naqvi http://orcid.org/0000-0002-9566-0153
Abhimanyu http://orcid.org/0000-0001-7888-8722
Gustavo Ramírez-Martínez http://orcid.org/0000-0001-5157-0677
Luis Jiménez-Alvarez http://orcid.org/0000-0002-5326-2720
Joaquin Zuniga http://orcid.org/0000-0002-7143-0281
Shabaana A Khader https://orcid.org/0000-0002-9545-4982

### Ethics

All human lung biopsy samples were obtained from the Tuberculosis Outpatient Clinic and the Department of Pathology at the National Institute of Respiratory Diseases (INER) in Mexico City, before TB treatment with informed consent, and with the approved protocol by the INER IRB for their use (project numbers B04-15 and B09-23). Also, lung samples from healthy controls (HC), non-TB individuals, were obtained from the tissue repository of the Department of Pathology at INER. No compensation was provided to the patients. The analysis was conducted at Washington University in St. Louis, School of Medicine, and approved by the University IRB (reference number 201811050).

Non-human primate procedures were approved by the Institutional Animal Care and Use Committee of Tulane National Primate Research Center and were performed following National Institutes of Health (NIH) guidelines. Male and female Indian rhesus macaques, verified to be free of Mtb infection by tuberculin skin test, were obtained from the Tulane National Primate Research Center. The animals were housed in an ABSL3 facility. All mice were maintained and used per the approved Institutional Animal Care and Use Committee (IACUC) guidelines at Washington University in St. Louis (IACUC approval 20190101) or University of Chicago (IACUC approval 72713).

Reviewer #1 (Public review): https://doi.org/10.7554/eLife.102634.3.sa1
Reviewer #2 (Public review): https://doi.org/10.7554/eLife.102634.3.sa2
Author response https://doi.org/10.7554/eLife.102634.3.sa3

## Additional files

### Supplementary files
MDAR checklist

Source data 1. Excel spreadsheets containing the meta data.

### Data availability
We did not generate new datasets and have used publicly available datasets mentioned in the section below.

The following previously published datasets were used:

| Author(s) | Year | Dataset title | Dataset URL | Database and Identifier |
|---|---|---|---|---|
| Esaulova E, Das S, Singh DK, Choreño-Parra JA, Swain A, Arthur L, Rangel-Moreno J, Ahmed M, Bucsan A, Moodley C, Mehra S, García-Latorre E, Zuniga J, Atkinson J, Kaushal D, Artyomov MN, Khader SA | 2021 | Defining the tuberculosis lung landscape during disease and latency using single cell technologies | https://www.ncbi.nlm.nih.gov/geo/query/acc.cgi?acc=GSE149758 | NCBI Gene Expression Omnibus, GSE149758 |
| Gideon HP, Hughes TK, Tzouanas CN, Wadsworth MH | 2022 | Multimodal profiling of lung granulomas in macaques reveals cellular correlates of tuberculosis control | https://www.ncbi.nlm.nih.gov/geo/query/acc.cgi?acc=GSE200151 | NCBI Gene Expression Omnibus, GSE200151 |

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
