## [Editor Report · eLife Assessment]

In this **useful** study, the authors utilize published scRNA-seq data to highlight the potential importance of mast cells (MCs) in TB granulomas, presenting a **solid** comparative assessment of chymase- and tryptase-expressing MCs in the lungs of *Mycobacterium tuberculosis*-infected individuals and non-human primates. While the authors appropriately discussed the inconsistencies across models, adoptive transfer experiments in MC-deficient mice would substantially strengthen the causal link between MCs and TB outcomes, providing more direct functional validation of the proposed role of MCs in TB pathogenesis.

---

## [Referee Report · Reviewer #1 (Public review)]

Summary:

The study by Gupta et al. investigates the role of mast cells (MCs) in tuberculosis (TB) by examining their accumulation in the lungs of *M. tuberculosis*-infected individuals, non-human primates, and mice. The authors suggest that MCs expressing chymase and tryptase contribute to the pathology of TB and influence bacterial burden, with MC-deficient mice showing reduced lung bacterial load and pathology.

Strengths:

The study addresses an important and novel topic, exploring the potential role of mast cells in TB pathology.

It incorporates data from multiple models, including human, non-human primates, and mice, providing a broad perspective on MC involvement in TB.

The finding that MC-deficient mice exhibit reduced lung bacterial burden is an interesting and potentially significant observation.

Results from a transfer experiment nicely substantiate the role of MCs in TB pathogenesis in mice.

---

## [Referee Report · Reviewer #2 (Public review)]

Summary:

The submitted manuscript aims to characterize the role of mast cells in TB granuloma. The manuscript reports heterogeneity in mast cell populations present within the granulomas of tuberculosis patients. With the help of previously published scRNAseq data, the authors identify transcriptional signatures associated with distinct subpopulations.

Strengths:

(1) The authors have carried out sufficient literature review to establish the background and significance of their study.

(2) The manuscript utilizes a mast cell-deficient mouse model, which demonstrates improved lung pathology during Mtb infection, suggesting mast cells as a potential novel target for developing host-directed therapies (HDT) against tuberculosis.

Weaknesses:

(1) The manuscript requires significant improvement, particularly in the clarity of the experimental design, as well as in the interpretation and discussion of the results. Enhanced focus on these areas will provide better coherence and understanding for the readers.

(2) The results discussed in the paper add only a slight novel aspect to the field of tuberculosis. While the authors have used multiple models to investigate the role of Mast cells in TB, majority of the results discussed in the Figure 1-2 are already known and are re-validation of previous literature.

(3) The claims made in the manuscript are only partially supported by the presented data. However, additional extensive experiments are necessary to strengthen the findings and enhance the overall scientific contribution of the work.

Comments on revisions:

While most of the comments have been addressed by the authors, a few important concerns pertaining to the data interpretation remain unanswered.

(1) The discrepancy between published studies and the current study on function of mast cells during TB remains. The authors could not justify the reason behind differences in results obtained during Mtb infection in humans vs macaques.

(2) To address the concern regarding immune alterations in mast cells deficient mice, the authors carried out adoptive transfer of mast cells to WT mice. However, they do not observe any changes in mycobacterial lung burden and inflammation, diluting their conclusions throughout the study.

(3) Additionally, as the authors propose mast cells as players in LTBI to PTB conversion, the adoptive transfer experiment could be conducted in a low-dosage model of TB. This would aid in assessing its role in TB reactivation.

---

## [Author Response]

The following is the authors’ response to the original reviews.

**Reviewer #1 (Public review):**
Summary:The study by Gupta et al. investigates the role of mast cells (MCs) in tuberculosis (TB) by examining their accumulation in the lungs of *M. tuberculosis*-infected individuals, non-human primates, and mice. The authors suggest that MCs expressing chymase and tryptase contribute to the pathology of TB and influence bacterial burden, with MC-deficient mice showing reduced lung bacterial load and pathology.Strengths:(1) The study addresses an important and novel topic, exploring the potential role of mast cells in TB pathology.(2) It incorporates data from multiple models, including human, non-human primates, and mice, providing a broad perspective on MC involvement in TB.(3) The finding that MC-deficient mice exhibit reduced lung bacterial burden is an interesting and potentially significant observation.Weaknesses:(1) The evidence is inconsistent across models, leading to divergent conclusions that weaken the overall impact of the study.

The strength of the study is the use of multiple models including mouse, nonhuman primate as well as human samples. The conclusions have now been refined to reflect the complexity of the disease and the use of multiple models.

(2) Key claims, such as MC-mediated cytokine responses and conversion of MC subtypes in granulomas, are not well-supported by the data presented.

To address the reviewer’ s comments we will carry out further experimentation to strengthen the link between MC subtypes and cytokine responses.

(3) Several figures are either contradictory or lack clarity, and important discrepancies, such as the differences between mouse and human data, are not adequately discussed.

We will further clarify the figures and streamline the discussions between the different models used in the study.

(4) Certain data and conclusions require further clarification or supporting evidence to be fully convincing.

We will either provide clarification or supporting evidence for some of the key conclusions in the paper.

**Reviewer #2 (Public review):**
Summary:The submitted manuscript aims to characterize the role of mast cells in TB granuloma. The manuscript reports heterogeneity in mast cell populations present within the granulomas of tuberculosis patients. With the help of previously published scRNAseq data, the authors identify transcriptional signatures associated with distinct subpopulations.Strengths:(1) The authors have carried out a sufficient literature review to establish the background and significance of their study.(2) The manuscript utilizes a mast cell-deficient mouse model, which demonstrates improved lung pathology during Mtb infection, suggesting mast cells as a potential novel target for developing host-directed therapies (HDT) against tuberculosis.Weaknesses:(1) The manuscript requires significant improvement, particularly in the clarity of the experimental design, as well as in the interpretation and discussion of the results. Enhanced focus on these areas will provide better coherence and understanding for the readers.

The strength of the study is the use of multiple models including mouse, nonhuman primate as well as human samples. The conclusions have now been refined to reflect the complexity of the disease and the use of multiple models.

(2) Throughout the manuscript, the authors have mislabelled the legends for WT B6 mice and mast cell-deficient mice. As a result, the discussion and claims made in relation to the data do not align with the corresponding graphs (Figure 1B, 3, 4, and S2). This discrepancy undermines the accuracy of the conclusions drawn from the results.

We apologize for the discrepancy which will be corrected in the revised manuscript

(3) The results discussed in the paper do not add a significant novel aspect to the field of tuberculosis, as the majority of the results discussed in Figure 1-2 are already known and are a re-validation of previous literature.

This is the first study which has used mouse, NHP and human TB samples from Mtb infection to characterize and validate the role of MC in TB. We believe the current study provides significant novel insights into the role of MC in TB.

(4) The claims made in the manuscript are only partially supported by the presented data. Additional extensive experiments are necessary to strengthen the findings and enhance the overall scientific contribution of the work.

We will either provide clarification or supporting evidence for some of the key conclusions in the paper.

**Reviewer #1 (Recommendations for the authors):**
In the study by Gupta et al., the authors report an accumulation of mast cells (MCs) expressing the proteases chymase and tryptase in the lungs of *M. tuberculosis*-infected individuals and non-human primates, as compared to healthy controls and latently infected individuals. They also MCs appear to play a pathological role in mice. Notably, MC-deficient mice show reduced lung bacterial burden and pathology during infection.While the topic is of interest, the study is overall quite preliminary, and many conclusions are not wellsupported by the presented data. The reliance on three different models, each suggesting divergent outcomes, weakens the ability to draw definitive conclusions. Specifically, the claim that "MCs (...) mediate cytokine responses to drive pathology and promote Mtb susceptibility and dissemination during TB" is not substantiated by the data.Major comments(1) In human samples, the authors conclude that "While MCTCs accumulated in early immature granulomas within TB lesions, MCCs accumulated in late granulomas in TB patients" and that MCTs "likely convert first to MCTCs in early granulomas before becoming MCCs in late mature granulomas with necrotic cores." However, Figure 1B shows the opposite. Furthermore, the assertion that MCTs "convert" into MCTCs is not justified by the data.

Corrections have been made to the figures to ensure clarity for the reader. We demonstrate accumulation of tryptase-expressing MCs in healthy individuals, while the dual tryptase and chymaseexpressing MCs were seen in early granulomas, and only chymase-associated MCs were observed in late granulomas depicting more pathology of the disease. We have removed the line as advised by the reviewer.

(2) In Figure 2 I and J, the panels do not demonstrate co-expression of chymase and tryptase in clusters 0, 1, and 3 in PTB samples, which contradicts the histology data. This discrepancy is left unaddressed and raises concerns about the conclusions drawn from Figures 1 and 2.

We thank the reviewer for pointing this out. We revisited the data and now show the coexpression of the dual expressing cells in the data (Figure 2H). This discrepancy stemmed from the crossspecies nature of the dataset. It turns out the there is a considerable diversity in sequence similarity and tryptase function between human and NHPs (Trivedi et al., 2007). We explain this in the section now (line 313-364). Briefly, while humans express TPSG1 (encoding tryptase) and TPSD1 (encoding tryptase) and have the same gene name in NHP, the gene name for more widely expressed TPSAB1(encoding / tryptase) is different for NHP and the gene names are not shared as they are still predicated putative protein. The putative genes from NHP that map to human TPSAB1 is LOC699599 for *M. mulatta* and LOC102139613 for M. fasicularis, respectively. Thus, looking for TPSAB1 gene yielded no result in our previous analysis but examining these orthologous gene names, now phenocopy the results we see in the histology data. To strengthen our findings, we have now analyzed an additional single-cell dataset from the lungs of NHP M. fasicularis (Figure 2J-L) and found the co-expression of chymase and tryptase, adding an important validation to our histological findings.

(3) Figure 2 serves more as a resource and contributes little to the core findings of the study. It might be better suited as supplementary material.

We thank the reviewer for the suggestion; however, we believe that Figure 2 serves as an independent validation in a different species (NHP), showing heterogeneity in MCs across species in a TB model. The figure adds value as there are only a handful of studies (Tauber et al., 2023, Derakhshan et al., 2022, Cildir et al., 2021) but none in TB, describing MCs at single cell level, of which one is published from our group showing MC cluster in Mtb infected macaques (Esaulova et al., 2021). We feel strongly that dissecting MCs as specifically done here provides an important insight into the transcriptional heterogeneity of these cells linked to disease states. We have also added an additional NHP lung single cell dataset (Gideon et al., 2022) to complement our analysis, thus adding another validation, strengthening these findings. So, we believe in retaining the figure as an integral part of the main paper.

(4) In lines 275-277, the data referenced should be shown to support the claims.

We thank the reviewer for the suggestion. The text originally noted by the reviewer now appears in the revised manuscript at line 370-372 and the corresponding data has now been included as supplementary Figure S3.

(5) In Figure 3B, the difference between the two mouse strains becomes non-significant by day 150 pi, weakening the overall conclusion that MCs contribute to the bacterial burden.

At 100 dpi, MC-deficient mice exhibit lower Mtb CFU in both the lung and spleen, indicating improved protection. By 150 dpi, lung CFU differences are no longer significant; however, dissemination to the spleen remains reduced in MC-deficient mice. Thus, the overall conclusion that MCs contribute to increased bacterial burden remains valid, particularly with respect to dissemination. This conclusion is further supported by new data showing that adoptive transfer of MCs into B6 Mtb-infected mice increased Mtb dissemination to the spleen (Figure 5E).

(6) Figures 3D and E are not particularly convincing.

Figures 3D and 3E illustrate lung inflammation in MC-deficient mice compared to wild-type which more distinctly show that MC-deficient mice exhibit significantly less inflammation at 150 dpi, supporting the role of MCs in driving lung.

(7) In Figures 4 and S3, the color coding in panels A-F appears incorrect but is accurate in G. This inconsistency is confusing.

We thank the reviewer for noting this. The color coding has been corrected to ensure consistency across all figures.

(8) In the mouse model, MCs seem to disappear during infection, in contrast to observations in human and macaque samples. This discrepancy is not discussed in the paper.

We thank the reviewer for this important observation. In response, we performed a new analysis of lung MCs at baseline in wild-type and MC-deficient mice. Our data show that naïve wild-type lungs contain a small population of MCs, which is further reduced in MC-deficient mice. Following Mtb infection, MCs progressively accumulate in wild-type mice, whereas this accumulation is significantly impaired in MC-deficient mice. These new data are now included in Figure (Figure 4A) and also updated in the text (line 395-403).

(9) In lines 306-307, data should be shown to support the claims.

We thank the reviewer for the suggestion. The text originally noted by the reviewer now appears in the revised manuscript at line 399-400 and the corresponding data has now been included as supplementary Figure S4.

Minor comments(1) What does "granuloma-associated" cells mean in samples from healthy controls?

We thank the reviewer for this point. The language has been revised to accurately refer to cells in the lung parenchyma in the Figure 1, rather than “granuloma associated” cells.

(2) In line 229, it is unclear what "these cells" refers to.

The phrase “these cells” refers to tryptase-expressing mast cells. This has now been clarified in the revised manuscript (line 276-277).

(3) The citation of Figure 3A in lines 284-285 is misplaced in the text and should be corrected.

The figure citation has been corrected in the text in the revised manuscript (lines 376-379).

**Reviewer #2 (Recommendations for the authors):**
(1) The data presented in Figure 1 seems to be a re-validation of the already known aspects of mast cells in TB granulomas. While distinct roles for mast cells in regulating Mtb infection have been reported, the manuscript appears to be a failed opportunity to characterize the transcriptional signatures of the distinct subsets and identify their role in previously reported processes towards controlling TB disease progression.

We thank the reviewer for the insight. While it was not our intent to investigate the bulk transcriptome, owing to the high number of cells required to get enough RNA for transcriptomic sequencing, it is technically challenging due to the low abundance of mast cells during TB infection (Figure 2). The motivation for Figure 2, that we utilized a more sensitive transcriptomic analysis to find the different transcriptional states in the distinct TB disease states. We believe that this analysis captures the essence of what the reviewer and provides meaningful insights into mast cell heterogeneity during TB.

(2) The experiments lack uniformity with respect to the strains of Mtb used for experimentation. For eg: Mtb strain HN878 was used for aerosol infection of mice while Mtb CDC1551 was used for macaques. If there were experimental constraints with respect to the choice, the same should be mentioned.

We thank the reviewer for this comment. The Mtb strain usage has been consistent within each species: HN878 for mice and CDC1551 for non-human primates (NHPs), in line with prior studies from our lab. The species-specific choice reflects the differences in pathogenicity of these strains in mice versus NHPs. CDC1551, which exhibits lower virulence, allows the development of a macaque model that recapitulates human latent to chronic TB when administered via aerosol at low to moderate doses (Kaushal et al., 2015; Sharan et al., 2021; Singh et al., 2025). In contrast, the more virulent HN878 strain leads to severe disease and high mortality in NHPs and is therefore not suitable for these models. Using CDC1551 in macaques provides a controlled and clinically relevant platform to study immunological and pathophysiological mechanisms of TB, justifying its use in the current study. This explanation has now been added to the manuscript method section (lines 109-114).

(3) Line 84- 85, the authors state that "Chymase positive MCs contribute to immune pathology and reduced Mtb control". Previous reports including Garcia-Rodriguez et al., 2021 associate high MCTCs with improved lung function. Additionally, in the macaques model of latent TB infection reported in the manuscript, the number of chymase-expressing MCs seems to significantly decrease. The authors should justify the same.

We thank the reviewer for this comment. In Garcia-Rodriguez et al., 2021, chymase-expressing MCs accumulate in fibrotic lung lesions. Fibrosis is a result of excessive inflammation in TB infection and is associated with lung damage. Similarly, in idiopathic pulmonary fibrosis, higher density and percentage of chymase-expressing MCs correlate positively with fibrosis severity (Andersson et al., 2011). In our study, although fibrosis was not directly assessed, chymase-positive MCs increased in late lung granulomas, consistent with advanced inflammatory disease. Therefore, our conclusion that chymaseproducing MCs contribute to lung pathology is justified and aligns with prior observations.

(4) The manuscript would benefit from a brief description of the experimental conditions for the previously published scRNAseq data used in the current study.

We thank the reviewer for the suggestion, and the information has been included in the final manuscript (lines 294-297) and represented as Figure 2A.

(5) The authors have not mentioned the criteria used to categorize early and late granulomas in TB patients. A lucid description of the same is necessary.

Based on reviewer’s comment the detailed categorization of early and late granulomas in TB patients is now included in the revised manuscript (line 256-260). Early granulomas: Discrete conglomerates of immune cells and resident stromal cells with defined borders and absence of central necrosis, and Late granulomas: Large and dense clusters of immune cells and resident cells with an evident necrotic center containing bacteria and dead neutrophils and lymphocytic infiltrating cells on the periphery of the necrotic center. MCs were measured in the periphery and inside early granulomas, while in the late granulomas, they were mainly quantified in the periphery.

(6) The authors mention that "While MCTCs accumulated in early immature granulomas within TB lesions, MCCs accumulated in late granulomas in TB patients". While this is evident from the representative, the quantification in Figure 1B seems to indicate otherwise.

We thank the reviewer for pointing this out. The labeling in the quantitative analysis shown in Figure 1B has been corrected in the revised manuscript to accurately reflect the accumulation of MC_TC_s in early granulomas and MC_C_s in late granulomas.

(7) The labelling followed in Figures 3, 4 and S2 do not match with the discussion. Such errors should be rectified to minimize any ambiguity within the text of the manuscript.

We thank the reviewer for noting this. The color coding has been corrected to ensure consistency across all figures.

(8) The mast cell deficient mice model has a differential number of immune cells at the site of granuloma as reported in the manuscript. This could contribute to the altered mycobacterial survival and inflammation cytokine production in the lung and hence might not be a direct effect of mast cell depletion. The authors can consider reconstituting mast cell populations to analyze the mast cell function.

We thank the reviewers for this suggestion. In the revised manuscript, we have adoptively transferred MCs into WT mice before Mtb challenge to assess if this would increase inflammation and Mtb CFU in the lung and spleen. Our results show that while lung inflammation was not impacted, we found that the dissemination to the spleen and the frequency of neutrophils in the lung were increased in WT mice that received MCs (Figure 5, lines 429-443).

(9) Line 295- 297, the authors state "MCs continued to accumulate in the lung up to 100 dpi in CgKitWsh mice, following which the MC numbers decreased at later stages". However, the quantification in Figure 4A does not reflect the same. This should be addressed.

In response to the reviewers' comments, we conducted a new analysis of lung MCs at baseline, comparing wild-type and MC-deficient mice. The revised data show that MC-deficient mice have fewer mast cells at baseline compared to B6 mice. Furthermore, mast cell numbers increase during infection, peaking at 100 days post-infection (dpi) and subsequently stabilize by 150 dpi. The revised data has been included in Figure 4A and text line 395-403.

(10) Additionally, while the scRNAseq data reflects a lower production of TNF in pulmonary TB granulomas, the mice deficient in mast cells are discussed to have a lower production of proinflammatory cytokines.

Mast cells increasing and contributing to the TB pathogenesis is the theme of the paper and as such we see and increase in the IFNG pathway genes and similar reduction in the production of pro- inflammatory cytokines. The relative decrease in the TNF pathway gene expression can be reconciled by the fact that less TNF gene expression in PTB could also represent loss of Mtb control and increased pathogenesis (Yuk et al., 2024), which is maintained in the LTBI/HC clusters. Higher bacterial burden of Mtb can also decrease the host TNF production, which is in line with what we observe here (Olsen et al., 2016, Reed et al., 2004, Kurtz et al., 2006).

(11) The authors have not annotated Figure 2 I and J in the text while describing their results and interpretation.

We thank the reviewer for noting this and the figure 2 has been revised and the results as pointed out have been added to the revised manuscript.

(12) In line 284, the authors have discussed the results pertaining to Figure 3B, however, mentioned it as Figure 3A in the text.

We thank the reviewer for noting this and the corrections have been made in the revised manuscript (lines 379-384).

References

ANDERSSON, C. K., ANDERSSON-SJOLAND, A., MORI, M., HALLGREN, O., PARDO, A., ERIKSSON, L., BJERMER, L., LOFDAHL, C. G., SELMAN, M., WESTERGREN-THORSSON, G. & ERJEFALT, J. S. 2011. Activated MCTC mast cells infiltrate diseased lung areas in cystic fibrosis and idiopathic pulmonary fibrosis. Respir Res, 12, 139.

CILDIR, G., YIP, K. H., PANT, H., TERGAONKAR, V., LOPEZ, A. F. & TUMES, D. J. 2021. Understanding mast cell heterogeneity at single cell resolution. Trends Immunol, 42, 523-535.

DERAKHSHAN, T., BOYCE, J. A. & DWYER, D. F. 2022. Defining mast cell differentiation and heterogeneity through single-cell transcriptomics analysis. J Allergy Clin Immunol, 150, 739-747.

ESAULOVA, E., DAS, S., SINGH, D. K., CHORENO-PARRA, J. A., SWAIN, A., ARTHUR, L., RANGEL-MORENO, J., AHMED, M., SINGH, B., GUPTA, A., FERNANDEZ-LOPEZ, L. A., DE LA LUZ GARCIA-HERNANDEZ, M., BUCSAN, A., MOODLEY, C., MEHRA, S., GARCIA-LATORRE, E., ZUNIGA, J., ATKINSON, J., KAUSHAL, D., ARTYOMOV, M. N. & KHADER, S. A. 2021. The immune landscape in tuberculosis reveals populations linked to disease and latency. Cell Host Microbe, 29, 165-178 e8.

GARCIA-RODRIGUEZ, K. M., BINI, E. I., GAMBOA-DOMINGUEZ, A., ESPITIA-PINZON, C. I., HUERTA-YEPEZ, S., BULFONE-PAUS, S. & HERNANDEZ-PANDO, R. 2021. Differential mast cell numbers and characteristics in human tuberculosis pulmonary lesions. Sci Rep, 11, 10687.

GIDEON, H. P., HUGHES, T. K., TZOUANAS, C. N., WADSWORTH, M. H., 2ND, TU, A. A., GIERAHN, T. M., PETERS, J. M., HOPKINS, F. F., WEI, J. R., KUMMERLOWE, C., GRANT, N. L., NARGAN, K., PHUAH, J. Y., BORISH, H. J., MAIELLO, P., WHITE, A. G., WINCHELL, C. G., NYQUIST, S. K., GANCHUA, S. K. C., MYERS, A., PATEL, K. V., AMEEL, C. L., COCHRAN, C. T., IBRAHIM, S., TOMKO, J. A., FRYE, L. J., ROSENBERG, J. M., SHIH, A., CHAO, M., KLEIN, E., SCANGA, C. A., ORDOVAS-MONTANES, J., BERGER, B., MATTILA, J. T., MADANSEIN, R., LOVE, J. C., LIN, P. L., LESLIE, A., BEHAR, S. M., BRYSON, B., FLYNN, J. L., FORTUNE, S. M. & SHALEK, A. K. 2022. Multimodal profiling of lung granulomas in macaques reveals cellular correlates of tuberculosis control. Immunity, 55, 827846 e10.

KAUSHAL, D., FOREMAN, T. W., GAUTAM, U. S., ALVAREZ, X., ADEKAMBI, T., RANGEL-MORENO, J., GOLDEN, N. A., JOHNSON, A. M., PHILLIPS, B. L., AHSAN, M. H., RUSSELL-LODRIGUE, K. E., DOYLE, L. A., ROY, C. J., DIDIER, P. J., BLANCHARD, J. L., RENGARAJAN, J., LACKNER, A. A., KHADER, S. A. & MEHRA, S. 2015. Mucosal vaccination with attenuated *Mycobacterium tuberculosis* induces strong central memory responses and protects against tuberculosis. Nat Commun, 6, 8533.

KURTZ, S., MCKINNON, K. P., RUNGE, M. S., TING, J. P. & BRAUNSTEIN, M. 2006. The SecA2 secretion factor of *Mycobacterium tuberculosis* promotes growth in macrophages and inhibits the host immune response. Infect Immun, 74, 6855-64.

OLSEN, A., CHEN, Y., JI, Q., ZHU, G., DE SILVA, A. D., VILCHEZE, C., WEISBROD, T., LI, W., XU, J., LARSEN, M., ZHANG, J., PORCELLI, S. A., JACOBS, W. R., JR. & CHAN, J. 2016. Targeting *Mycobacterium tuberculosis* Tumor Necrosis Factor Alpha-Downregulating Genes for the Development of Antituberculous Vaccines. mBio, 7.

REED, M. B., DOMENECH, P., MANCA, C., SU, H., BARCZAK, A. K., KREISWIRTH, B. N., KAPLAN, G. & BARRY, C. E., 3RD 2004. A glycolipid of hypervirulent tuberculosis strains that inhibits the innate immune response. Nature, 431, 84-7.

SHARAN, R., SINGH, D. K., RENGARAJAN, J. & KAUSHAL, D. 2021. Characterizing Early T Cell Responses in Nonhuman Primate Model of Tuberculosis. Front Immunol, 12, 706723.

SINGH, D. K., AHMED, M., AKTER, S., SHIVANNA, V., BUCSAN, A. N., MISHRA, A., GOLDEN, N. A., DIDIER, P. J., DOYLE, L. A., HALL-URSONE, S., ROY, C. J., ARORA, G., DICK, E. J., JR., JAGANNATH, C., MEHRA, S., KHADER, S. A. & KAUSHAL, D. 2025. Prevention of tuberculosis in cynomolgus macaques by an attenuated *Mycobacterium tuberculosis* vaccine candidate. Nat Commun, 16, 1957.

TAUBER, M., BASSO, L., MARTIN, J., BOSTAN, L., PINTO, M. M., THIERRY, G. R., HOUMADI, R., SERHAN, N., LOSTE, A., BLERIOT, C., KAMPHUIS, J. B. J., GRUJIC, M., KJELLEN, L., PEJLER, G., PAUL, C., DONG, X., GALLI, S. J., REBER, L. L., GINHOUX, F., BAJENOFF, M., GENTEK, R. & GAUDENZIO, N. 2023. Landscape of mast cell populations across organs in mice and humans. J Exp Med, 220.

TRIVEDI, N. N., TONG, Q., RAMAN, K., BHAGWANDIN, V. J. & CAUGHEY, G. H. 2007. Mast cell alpha and beta tryptases changed rapidly during primate speciation and evolved from gamma-like transmembrane peptidases in ancestral vertebrates. J Immunol, 179, 6072-9.

YUK, J. M., KIM, J. K., KIM, I. S. & JO, E. K. 2024. TNF in Human Tuberculosis: A Double-Edged Sword. Immune Netw, 24, e4.